# Chemistry of Dimer Acid Production from Fatty Acids and the Structure–Property Relationships of Polyamides Made from These Dimer Acids

**DOI:** 10.3390/polym15163345

**Published:** 2023-08-09

**Authors:** Charles R. Frihart

**Affiliations:** Forest Products Laboratory, USDA Forest Service, One Gifford Pinchot Drive, Madison, WI 53726-2398, USA; charles.r.frihart@usda.gov; Tel.: +1-608-231-208

**Keywords:** fatty acids, dimerization, clay catalyst, dimer acid, isostearic acid, polyamide, hot melt, structure–property relationships, gels

## Abstract

While there is abundant literature on using a wide range of biomaterials to make polymers for various adhesive applications, most researchers have generally overlooked developing new adhesives from commercially available bio-based dimerized fatty acids. Some of the literature on the chemistry taking place during the clay-catalyzed dimerization of unsaturated fatty acids is generally misleading in that the mechanisms are not consistent with the structures of these dimers and a by-product isostearic acid. A selective acid-catalyzed interlayer model is much more logical than the widely accepted model of clay-catalyzed Diels–Alder reactions. The resulting dimers have a variety of linkages limiting large crystal formation either as oligomeric amides or polyamides. These highly aliphatic fatty acid dimers are used to make a wide range of hot melt polyamide adhesives. The specific structures and amounts of the diacids and diamines and their relative ratios have a big effect on the bio-based polyamide mechanical properties, but analysis of the structure–property relationships has seldom been attempted, since the data are mainly in the patent literature. The diacids derived from plant oils are valuable for making polyamides because of their very high bio-based content and highly tunable properties.

## 1. Introduction

The interest in making adhesives more bio-based has increased in recent years, but unfortunately much of the recent literature has been directed at using materials that have limited commercial viability due to the small volumes available and the complexity of isolating the valuable bio-based materials. Many of the recent reviews [1,2,3,4,5], except for one [6], have given little attention to the wide use of fatty acid (FA) bio-based materials for industrial applications, except as bio-fuels and epoxidized fatty oils; the fascinating chemistry involving technology developed in the mid-20th century for making polyamide adhesives has received only limited mention. The early work on the dimerization process had been thoroughly reviewed many years ago [7,8,9,10], and with most of the polyamide literature in patents, the public domain knowledge is mainly related to performance properties. This paper concentrates on the interesting chemistry for converting the monomeric FA into dimer and oligomeric acids using clay catalysts and then the structure–property relationships in using these dimers for making a wide variety of commercial oligomeric products and polyamide hot melt adhesives.

While the main source of fatty esters is agricultural crops (e.g., oilseeds), fatty esters are present in all plants. Most of these esters are consumed as foods, but now there is a growing interest in trans-esterifying them for bio-based liquid fuels and modifying them as partial replacement for petroleum-based monomers used in making polymers. Fatty esters and their hydrolysis products FA have long been used in many industrial applications, including coatings, surfactants, and adhesives [11,12,13,14,15].

The interest in evaluating the polymerization of fatty esters was derived from research on improving drying oil coatings, which involve the free radical oxidation of polyunsaturated fatty esters with air exposure after incorporation of certain metal salts [16,17]. Thermal polymerization of polyunsaturated fatty esters in the absence of air was shown to provide non-oxidative intermolecular reactions at the olefinic sites, and after hydrolysis of these dimer esters, the resulting dicarboxylic acids were used to synthesize polyamides [18]. However, investigation into converting the polyunsaturated FA into polymeric materials by a thermal process was hampered by some loss of the carboxylic acid functionality and increase in color. This problem was solved by Goebel, who added water to the thermal heating step [19]. Later on, a process using a montmorillonite clay catalyst to lower the reaction temperature proved to be the best route for commercial production of a product with a high diacid content and lighter colors not only from polyunsaturated FA, but also from the monounsaturated FA oleic acid [8,20,21].

The literature on the function of a clay catalyst is not always clear, which has led to confusion in various publications on the reaction mechanism and the products from the process [10,22,23,24]. Thus, the first objective of this paper is to clarify the role of the clay and how it leads to unique monomeric, dimeric and polymeric acids. The second objective is to provide an understanding on how these unusual dimers lead to the observed structure–property relationships of associative oligomers and hot melt polyamides. For clarity, instead of calling the initial dimer acid and higher polymers fraction dimer acids, this paper uses the commonly used term polymerized FA to keep it distinct from the purified dimer fraction. Because of the interest in adhesives with a high bio-based content, the lack of recent reviews on the dimerization process and the polyamide adhesives produced from these dimer acids, with over 90% bio-based content, is surprising. Hopefully, an understanding of the dimerization process and the properties of the dimer polyamides will lead others to do further research in these areas.

## 2. Fatty Acid Dimer Synthesis and Properties

### 2.1. Clay Structure and Property

Natural and modified clays have long been used in the chemical industry [25]. A very important use in the fatty oils industry is as bleaching earths, involving clays such as kaolin, bentonite, sepiolite or attapulgite. These clays can absorb all kinds of substances, such as oil, insecticide, and hydrocarbons [25,26]. Activated bleaching earths are often used for bleaching vegetable oils to remove colored and distasteful components for food uses.

Most clays are layered alumino-silicates composed mainly of fused layers of tetrahedrally arranged silicate and octahedrally arranged aluminate groups [25,26,27]—see Figure 1—and many are known to be acidic catalysts. Clays are mainly stacked sheets, with each sheet composed of an aluminate layer in between two silicate layers [25,26]. All the clays have a negative surface charge due to interruption of sheet structure at the edge of the particles. These negative lattice surface charges are balanced by adsorbed metal cations as shown in Figure 1. These sheet structures are interesting in that the kaolinites, with a formula of Al_2_Si_2_O_5_(OH)_4_, are virtually non-swelling, since water cannot overcome the affinity between the sheets and be absorbed between them, as illustrated in the left side structure in Figure 1. On the other hand, the swelling smectite group including talc, vermiculite, montmorillonite, and bentonite have negative charges within the lattices with a general formula of (Ca,Na,H)(Al,Mg,Fe,Zn)_2_(Si, Al)_4_O_10_(OH)_2_-xH_2_O, where x represents a variable amount of water. Replacing trivalent aluminum with divalent atoms and replacing the tetravalent silicon with trivalent or divalent atoms leads to a lattice that is highly negatively charged as illustrated on the right side of Figure 1; the charges are balanced by cations between the sheets as well as on the particle surface. This propping of sheets allows water and fatty acids to flow in between the sheets. The amount and type of cations influences this swelling character of the montmorillonites [27]. An important property of clays is their cation exchange capacities of 3–15 milliequivalents per 100 g for non-swelling kaolinites versus 80–120 for swelling smectites, which is also related to their surface areas of 5–40 square meters per gram for kaolinites versus 40–800 for smectites and swelling abilities with an intersheet spacing of 7 Ǻ for kaolinites versus 9.6–20 for smectites [26]. Because of the charge separation between the modified aluminate-silicate lattice and the adsorbed cations, these cations give the clays their acidity [25,26,27]. Thus, the smectite clays not only have many active cations with a large surface area but have expandable intersheet separations to allow the unsaturated FA to be adsorbed and converted to carbonium ions as the intermediate for dimerization.

To avoid confusion with highly activated clays, these dimerization clay catalysts do not have the replacement of the metal cations by hydrogen atoms and are not subjected to the very high temperature calcining step that provide very reactive Lewis acid clays with collapsed layers and that along with zeolites are used in petroleum cracking [28]. Another point is that these clays still contain the stacked sheet structures and are not the exfoliated clays [29] which often are treated with bulky amine groups to separate the clay sheets, as illustrated by the bottom left structure on Figure 1 [30]. However, even without going to the extent of exfoliation, fatty acids (FA) can intercalate between the sheets of smectite clays, as illustrated in the structure on the right side of Figure 1 [8,31]. This good affinity of the clays for fatty acids leads to one drawback for the filtration process in that the residual clay contains an equal weight of fatty acids and is a disposal challenge because oxygen exposure can lead to autoignition from uncontrolled oxidation of the residual FAs.

The process of adding clay to the unsaturated FA and heating results in the FA penetrating the intersheet domains of the smectite particles [31,32] and being near the numerous acidic cations in a confined environment supporting reactions of the FA olefinic portion [8,31,33]. Layered clays allow the reaction products (dimer and polymers) to escape from the reaction site, which may not be true for zeolites, and be replaced by fresh unsaturated fatty acids. Commercial production uses the montmorillonite/bentonite clay and lower temperatures of approximately 250 °C compared to the 300 °C of the thermal process without clays [10], along with lithium hydroxide addition to reduce color and increase the conversion to dimers [23,34].

### 2.2. Dimerization Products and Structures

A typical procedure [35] uses TOFA (tall oil fatty acids, with a composition of 48.8% oleic acid, 34.3% linoleic acid, 6.4% conjugated linoleic acid, and 8.5% saturated C2–C20 acids), 4.3% montmorillonite clay, 1.1 mg of lithium salt per gram of clay, and 5% water stirred and heated in an autoclave at 260 °C. and 90 psi for 2.5 h. The crude product mixture is treated with 1.1% of 85% phosphoric acid at 130 °C, and then filtered to remove the catalyst. The yield of residual polymerized FA is commercially 63% recovered after vacuum distillation using a wiped film evaporator, which creates a hot thin film of fatty acid being distributed by a rotating blade wiping the interior of the heated cylinder under vacuum, allowing the evaporation of the more volatile monomeric FA (Figure 2). The polymerized FA can then be redistilled at higher temperature and lower vacuum pressure to yield a purified dimer [8,11] along with the residual dimer, trimer, and higher-molecular-weight products.

In the first distillation, the monomers are removed from the product mix. Instead of being just unreacted linear unsaturated FA, the majority are the branched isoacids [11]. These isoacids are isolated by hydrogenation in some cases and removal of the solid stearic acid, which melts at 69 °C. The resulting saturated isoacids are of interest because they are liquid, resistant to oxidation, and can be used in cosmetics, lubricants, and biodiesel fuel [11,36] in contrast to the liquid unsaturated fatty acids, which are sensitive to oxidative degradation, and to the saturated FA that are solids at room temperature. Some recent research has involved rearrangement of fatty esters or acids using acidic zeolites to produce isoacids [36]. The isostearic acids from the clay-catalyzed reaction involve rearranging the fatty acid chain to create methyl side groups, mainly in the middle of the chain near the sites of the olefinic sites in the feedstock; this reaction is favored since it produces the more stable tertiary carbonium ion instead of the first formed secondary one [34,37,38]. Thus, the liquid nature and oxidative resistance of the saturated isoacids is their main market advantage [39]. There are some natural isoacids, but they are low in concentration in the plant material. There have been times when the by-product isostearic acid has been more valuable than the dimer acids, which led to investigation into ways to improve their yield [37].

There are a variety of commercial polymerized fatty acids since there are many different markets for these products, and the use of oleic acid from soybeans instead of TOFA from wood pulping provides different product ratios [8,11,40]. The polymerized FA contains dimer, trimer, and higher-molecular-weight products with the latter two being useful to increase the functional groups per molecule that are important for applications such as amidoamines used in epoxy curing agents. However, for many hot melt polyamide products, which is one focus of this paper, the dimer is the preferred product since it can be used to make higher-molecular-weight polymers without gelling. Although the TOFA is the preferred feedstock for many companies making polyamides, for some hot melt polyamides oleic acid is preferred to make polyamides where their light color and stability after hydrogenation are important. Hydrogenation is difficult for TOFA-derived dimer due to impurities that tend to deactivate the hydrogenation catalysts.

Characterization of the reaction products has required several analytical methods. The data indicated that clay-catalyzed oleic dimer consisted chiefly of unsaturated non-cyclic and monocyclic non-aromatic dimer structures; see Figure 3 and Table 1 [38]. However, the more unsaturated FA monomers produced more aromatic and polycyclic dimers. The literature can be confusing in that for the earlier work, determining the structures in the complex reaction mixture was difficult since sophisticated analytical methods were not available to these researchers. The analysis is also complicated due to formation of lactones, hydroxyl FA, and inter-esters by addition of the carboxylic acid across the olefinic bond, in addition to the branched monomers, and dimers, trimers, and higher-molecular-weight products with different structures [22,38,41]. Even though distillation methods are difficult because of the high molecular weights and viscosity of the components, distillations can separate monomeric forms of dimeric and polymeric products, even using wiped-film evaporators. The purity of these fractions can be assessed using gel permeation chromatography [38].

The structure of the dimers has been studied in detail, using various chromatographic separations, nuclear magnetic resonance, ultraviolet, and mass spectroscopy for the molecular weight determination of the various components [38]. The molecular ion region of the mass spectrum was used to determine the number of ring systems and double bonds present in dimer structures. The confounding effect of olefinic bonds was eliminated by hydrogenation using conditions that eliminated only the olefinic bonds and not the aromatic ones, and methyl esterification was used to make the molecules more volatile. Thus, the non-cyclic (linear), monocyclic, bicyclic, and aromatic components were measured. From the fragmentation pattern of the mass spectrum, the lengths of the side chains attached to the ring systems were determined. Because mass spectroscopy is not quantitative, the aromatic content was determined by using ultraviolet spectroscopy, while nuclear magnetic spectroscopy of the non-hydrogenated and hydrogenated dimer was used to estimate the olefinic content of the dimer. With these experiments and other literature, McMahon and Crowell proposed structures in Figure 3 as representative of the dimer components [38]. Because of the variety of unsaturated fatty acids from natural sources and the multiple olefinic bond locations, these structures can only be representative and not the only isomers that exist, with the amount of unsaturation influencing the distribution (Table 1).

The data clearly indicate that, as expected, the oleic dimer which is high in mono-unsaturation, provides almost as much linear dimer as it does the non-aromatic monocyclic dimer. The data show that den Otter’s assumption regarding the high hydrogen transfer and Diels–Alder cyclization as the main pathway for dimer formation with oleic acid [23,42] is not a valid assumption because of a high percentage of non-cyclic (linear) dimer. TOFA containing both mono- and poly-unsaturation has values between the oleic acid and the highly unsaturated linoleic acid leading to a higher content of non-aromatic monocyclic dimer. The dimer and isoacid structures are important for understanding the dimerization mechanism, which is discussed in the next section, and for knowing the structure–property relationships of polyamides.

The variety of structures and the long flexible backbone make the dimer derivatives less likely to form large crystals compared to the common nylon structural polymers [43], but such as with structural nylons, the strong hydrogen bonds play an important role in the polymer strand associations [9]. A clear example of this is in making a resin for use in a hot melt ink jet printer that needs to be low in viscosity to be jetted by a printer head, clear to have sufficient color intensity, and hard to be rub resistant, as well as bonding sufficiently to plastic and coated papers. The composition has a narrow window to meet the requirements of this application, with the target oligomeric structure illustrated in Figure 4. Reacting more than the stoichiometric amount of stearic acid leads to cloudiness due to stearic acid chain crystals, while less leads to higher molecular weight and higher viscosity [44,45]. The successful associative oligomer probably has small crystallites that are less than the wavelength of normal light leading to clarity, but the insolubility and hardness of the product can be explained by the product having a high association of the oligomers through hydrogen bonding of the amide groups. This associative oligomer along with a compatible viscosity reducer for a single melt transition allowed the combination to be used commercially for high-end inkjet printers that even worked with glossy surfaces.

In contrast to the high hardness and very low solubility of the hot melt jet printing ink oligomer, other associative dimer-based oligomers have high compatibility with non-polar solvents such as mineral oil. These oligomers link two dimer units with a short chain diamine and then the terminal carboxylic acid is capped by reaction with a fatty alcohol, such as stearyl alcohol to provide non-polar domains; see Figure 5. These molecules use the good intermolecular hydrogen bonding ability of the center amides to provide structure for the gels while the low polarity, terminal esters provide compatibility with the mineral oil. After proving commercial success with clear scented candles [46,47], other proprietary formulations were developed for a variety of consumer products due to the presence of both polar and non-polar domains involving the individual oligomeric chains.

The two associative oligomer structures mentioned above show that the polar amide groups can provide strength, while the long terminal fatty chain dimer provides association of hydrophobic domains. Large crystalline domains do not form due to the various cyclic non-aromatic, cyclical aromatic, and acyclic domains inhibiting an orderly packing.

### 2.3. Dimerization Mechanism

From the earliest days of dimerization, an acidic process has been considered important, but the literature proposes other mechanisms. The most recited mechanism has been a Diels–Alder reaction to form six member rings [23,48] with den Otter proposing that this was the main route even for the mono-unsaturated oleic acid. The formation of six-member rings were supported by early analysis of some of the dimer molecules [22], but the Diels–Alder cyclization mechanism is not valid for fatty acid dimerization. The well-studied and modelled Diels–Alder reaction involves a concerted cyclization through a single, cyclic transition state, without any intermediates being produced along the way [49,50]. The model requires appropriate molecular overlap between a conjugated diene and an olefin, called a dienophile, with the latter being not just any olefin, but one that needs to be bonded to an electron withdrawing group. The mid-chain olefinic bond in the unsaturated FAs does not fit this description since it is not directly attached to a strong electron withdrawing group. To fit his model for oleic acid dimerization, den Otter mentioned that the Diels–Alder reaction can be acid catalyzed and assumed that this is the reason for the effectiveness of the clay [48]. However, the literature on acid catalysis of Diels–Alder reactions indicates that the catalyst needs to be a Lewis acid type associating with the electron withdrawing group attached to the dienophile [50,51]. This clay-catalyzed dimerization involves Bronsted acids, not a Lewis acid since they are used under aqueous conditions. In contrast, carbonium ion reactions explain both the observed formation of isoacids and the acyclic dimer [51,52], which the den Otter Diels–Alder mechanism does not. A conjugation diene can still cyclize with an olefin in a multi-step reaction carbonium ion process, as opposed to the single stage Diels–Alder reaction. Thus, the extensive reference to a Diels–Alder cyclization in the literature is not correct. The den Otter mechanism is also incorrect since it requires extensive hydrogen transfer between molecules and does not account for a non-cyclic dimer being a major component of the product and for the formation of isoacids [38].

The metallic or hydrogen cations can generate carbonium ions with the olefinic portion of unsaturated acids, leading to coupling of olefinic bonds, olefinic bond isomerization and conjugation, as illustrated in Figure 6 [51,52]. The stability of tertiary carbonium ions explains forming methyl side chains of the isostearic acid [51]. The carbonium ion can extract a hydrogen adjacent to an olefinic bond on another FA to produce a more stable carbonium ion due to resonance stabilization of the charge. Loss of a proton on the carbon adjacent to the carbonium ion leads to a conjugated olefin and a more reactive FA for further reaction, and the hydrogen transfer can then lead to the observed aromatic dimers [38]. Although den Otter did extensive analysis of the reaction of the monounsaturated oleic acid, his modelling was predicated on the dimerization being exclusively through a rapid Diels–Alder reaction for which approximately half of the oleic acid would have to be converted first to a diunsaturated linolenic acid in a slow hydrogen transfer step to provide the necessary diene for forming exclusively cyclic dimers. Not only does the cyclic dimer does not need to be formed by a concerted reaction Diels Alder reaction [23,42], but also the mass spectroscopy work shows that the predominate dimer is not a cyclic structure [38]. This unfortunate assumption renders all his systematic analysis moot.

Among the by-products of the dimerization reaction is the unwanted formation of inter-esters, also referred to as an estolide, that have been discussed even in the first patents on the dimerization process using a clay catalyst [20,21,41]. This reaction involves the addition of the carboxylic acid of one FA across the olefinic bond of another FA molecule. This type of reaction has been studied in a simpler system using a strongly acidic ion exchange resin [53] and modifying the clay catalyst to enhance desired production [41]. This process also occurs intra-molecularly to form a lactone, as well as inter-molecularly to form inter-ester. The easiest way to identify these side reactions is to measure the difference between the acid and saponification numbers [19]. Both inter-esters and lactones are problematic for making hot melt polyamides since their mono functionality serves as chain terminators limiting the polyamide molecular weight. Another issue is the potential of decarboxylation reducing the functionality of dimers and trimers based on the literature reports [19,54]. However, other literature has indicated that the decarboxylation requires higher temperatures and specific catalysts [55]. The clay-catalyzed dimerization minimizes these side products compared to strong acid catalysts such as sulfuric acid, probably by the counter ions forming salts of the fatty acids at the reaction site in the clay, and therefore is the best way to make dimer acids.

## 3. Dimer-Based Polyamides

### Polyamide Composition and Properties

From the beginning of the dimerization of FA technology, the importance of converting these dimers to polyamides was realized as being an important outlet for FA [19]. Dimer-based amides are used in applications such as adhesives, printing inks and coatings, because of their adhesion to both polar and non-polar surfaces and their good strength while maintaining flexibility [9,22,24,39]. While one broad usage is as a curing agent for epoxy adhesive [10,19], this paper focuses on its use for hot-melt adhesives due to the limited number of papers or patents relating structure–property relationships to end use applications. This is especially true in how dimer-based amides differ from the more common nylons in being less crystalline and more hydrophobic, but also being similar to nylons in how specific hydrogen bonds of the amides play a large role in the polymer properties.

Typical nylons have good resistance to most organics due to the strong hydrogen bonding, but this comes at the cost of high moisture absorption relative to most thermoplastics. The amide groups made from primary amines provide very good reversible cross links through hydrogen bonds with the amides being both a proton donor and acceptor [43]. The regular structure of the homo-polymers leads to good crystallinity, although it should be noted that even chain length monomers have higher melting points than odd chain length monomers due to the difference in crystal structures, and as the number of methylene groups increases between the amide groups, the melting point also declines; see Table 2. The density of amide groups and the even versus odd chain lengths establish the nature of the crystallinity, which has a profound effect on the strength and heat resistance of the dimer polyamides. Thus, it is not surprising that with dimer acid having 34 carbon atoms between carboxyl groups, the softening point is much lower than the melting points of structural nylons; for dimer polyamides the softening point is measured since they do not have a distinct melting point. In addition, with the dimers having a variety of mono- and bi-cyclic along with non-cyclic structures, the ability to form large, distinct crystalline structures is unlikely. The hydrogen bonding structure alters properties other than melting/softening points. As the length of the diacids and/or diamines increases, the stiffness drops, but the water resistance and elongation improve.

While the hydrogen bonds play a key role in the dimer-based polyamides, the dimer structure also has a large effect on the polyamide properties [9,22,24,57]. As expected, the 34 carbons between the amide groups makes the product less hydrophilic and more hydrophobic than the 4 carbons in nylon 6,6. Even with the shortest diamine, the dimer polyamide has a low softening point (100 °C), which is not desirable for many hot-melt adhesive applications and is likely to have a creep problem. This drawback of reduced hydrogen bonding is even more serious by extending the length of the diamine portion with hexamethylenediamine and dimer diamine; see Table 2. Consequently, replacing part of the dimer acid with shorter chain diacids, such as the bio-based sebacic or azelaic acids, provides improvement in the softening point along with higher tensile strength, at only a modest reduction in elongation; see Table 3 [24,58]. The increase in sebacic acid amounts improved the softening point of the product, with just 5% replacement resulting in a 35 °C increase in the softening point and a 150 MPa increase in tensile strength, while reducing the elongation by 100%, emphasizing how important hydrogen bonding is to the overall properties of the polyamides.

As mentioned, using the co-diacids sebacic and azelaic acids keep these polyamides as highly bio-based materials. The commercial natural product ricinoleic acid (12-hydroxy-9-cis-octadecenoic acid) is isolated from castor oil, and then subjected to caustic fusion to produce sebacic acid [59,60]. Although this is a truly bio-based compound, the use of lead oxide in some cases to improve the product yield in the oxidation reaction makes this material less green. On the other hand, the azelaic acid is made by oxidative ozonolysis of oleic acid in a more environmentally friendly process and produces useful pelargonic acid as the main by-product [59,61]. Most diamines are not bio-based, but they are generally a small weight percent of the various dimer polyamide formulations.

The literature does not discuss the difference in improvement of properties when using azelaic compared to sebacic acid. However, the use of 20- or 18-carbon linear dibasic acid as the copolymerizing diacid with a polymeric fatty acid and various diamines to prepare polyamide hot-melt adhesives provides resins with better tensile strengths at ambient and elevated temperatures and increased moisture resistance than with the sebacic or azelaic acids [62]. It was proposed that these property improvements are derived primarily from the increased crystallinity imparted to the polyamides by the long chain, linear 20- or 18-carbon dibasic acid by incorporating the dimer into the co-diacid crystallites since they are more similar in chain lengths than with the dimer than the sebacic acid (10 carbons) and the azelaic acid (9 carbons). These much more expensive co-diacids are available commercially and are bio-based, but the improvement has not been cost effective because of the ability to form a wide range of properties with the conventual monomers [8,10].

These standard polyamides of dimer, co-diacid, and ethylenediamine or hexamethylenediamine are useful in many industrial applications [8,11,22,24,56,63] due to their good bonding to dissimilar substrates, resistance to water, oil and greases, and little softening of the polyamide until the softening point is reached. Most other hot melt adhesives, such as polyethylene, polypropylene, ethylene-vinyl acetate, and polyurethane generally cannot compete dimer acid polyamides in higher performance applications [24,64]. These other polymers depend on entanglement and van der Walls forces, rather than the reversible cross linking through hydrogen bonds. The wide range of properties of dimer acid, hot melt polyamide adhesives is controlled by the diacid and diamine formulations, which are mainly industrial secrets, but not by compounding with additives that can migrate with time to cause bond deterioration through loss of interface strength [24]. However, non-bio-based, moisture-cured hot-melt polyurethanes developed in recent years with good high temperature and moisture resistance properties have provided competition to the dimer polyamides in some markets [65].

Over the years, use of other diamines has brought greater utility to the polyamides for other important markets. A key example is that the standard polyamides have little adhesion to plastics, including polyvinyl chloride. Replacing some of the shorter chain diamines with piperazine (Figure 7) results in adhesion to polyvinyl chloride and other plastics [66]. This improvement in plastic bonding was not explained until Frihart provide an explanation for the role of piperazine [67]. After showing that rheology and solubility parameter models of piperazine-containing polyamides were inconsistent with the excellent performance of the piperazine-containing dimer polyamides compared to those without piperazine; the acid–base interaction model was shown to be consistent with the data in the literature [58,64,67,68,69,70]. The amide made from piperazine with its secondary amines is only a proton acceptor compared to amides made from the typical primary amines despite the increased rigidity of the individual diamide bonds given the lower flexibility of the six-member ring. Even polyamides made from dimer and dimerdiamine do not bond vinyl despite the limited number of hydrogen bonds. Increasing the amount of piperazine with respect to ethylenediamine led to decreased rate- and time-dependent shear thinning rheometry indicating that the increased piperazine amount led to decreased interchain hydrogen bonding [64]. Thus, the piperazine-containing polyamides have sufficient acceptor sites not tied up in internal cohesion bonds so that many are available for externally bonding to the acidic proton on polyvinyl chloride, while most polyamides made from just primary amines have their acceptor sites mainly tied up in internal hydrogen bonds [67]. Another approach is to use diols or amino-alcohols to provide mode flexibility and bonding to plastics [71,72] due to the alcohol only forming hydrogen-acceptor esters, but not hydrogen-donating bonds.

Even though the amide groups are typically approximately 15% by weight of the polyamide, they show a strong role in the polyamide properties. The crucial role that the diamines play in the polyamide properties is revealed by comparison of 1,2-diaminopropane (DAP, Figure 7) and 1,3-daimopropane or ethylenediamine (EDA) [68]. EDA is a standard diamine component used to obtain strong fast setting polyamides, most likely due to good ability to form hydrogen bonds that lock up polymer segments due to the very limited rotation of the diamide structures. However, seemingly small changes in the diamines make a big difference in the polyamide properties, see Table 4. The DAP, which has an even chain length between amide groups such as ethylenediamine, provides polyamides with reasonable tensile strength and modulus, while the 1,3-diaminopropane, which has an odd chain length between amide groups, never develops a truly solid material since it slowly cold flows at ambient conditions; see Figure 4. The poor performance of the odd chain length diamines is reinforced by the example of 2-methyl-1,5-pentanediamine giving a polyamide with cold flow problems [68], while its isomer hexamethylene diamine does not. An interesting property is that the more hindered DAP has a longer open time allowing for better positioning of the substrates before the adhesive hardens in contrast to the fast-setting ethylenediamine but still forms a polyamide with good strength despite the hinderance of the methyl group near one of the amine groups. Thus, the extra methyl group hinders bond rotation but does not prevent the formation of intermolecular hydrogen bonds, while the more flexible 1,3-dimainopropane in these formulations, also containing piperazine, does not establish strong hydrogen bonding domains, most likely preventing formation of substantial crystallites.

Another change in the diamine part of the polyamide formulation is to use diamines with polyether groups to improve impact resistance and low temperature properties [70]. The polyether domains provide low glass transition temperature due to the reduce hinderance to rotation compared to hydrocarbon domains.

Traditional markets for these adhesives are in footwear, cabinet assembly, multiwall bag closure, vinyl-clad windows, heat shrinkable telecommunication cable connectors, and other assembly applications in packaging, automotive, and electrical industries [22,24]. The commercial advantage of the hot melt polyamides is that the amide bonds retain their structure until near the softening point followed by rapid viscosity drop for ease of application and then fast strength development when the heat is removed. The combination of high aliphatic bio-based content and the strong amide bonds led to wide chemical and grease resistance, along with good heat resistance.

Unfortunately, the literature is lacking some data that can provide an improved understanding of the effect of crystallinity on polyamide performance, with a notable example being dynamic mechanical analysis sweeps with increasing temperature. These data could provide the glass transition temperature, and softening point transitions. and loss of rigidity with increasing temperature. Related bio-based research areas that have been mainly ignored are naval stores, which includes other TOFA adhesive uses [14] and TOFA based amine hardeners for epoxies [15].

## 4. Conclusions

The dimerization of unsaturated fatty acids with a clay catalyst, and conversion of these to polyamide adhesives, have been commercialized for over half a century, but a critical examination of the chemistry involved in both processes has not been published. Some confusion over the structure of dimer products has been resolved by improved analytical methods [38]. Many places in the literature wrongly describe the dimerization process as occurring by a Diels–Alder reaction, which needs a dienophile activated by an attached electron withdrawing group, which is not present in this case. The clays are known to be acidic catalysts and the montmorillonites used in this process are swelling clays that allow for a confined acidic reaction environment [8,10]. This type of reaction is also consistent with the branched monomers and the variety of dimer and higher polymer structures produced in dimerization.

Although the original polymerized fatty acids find many uses, the isolation of the dimer fraction results in a variety of amides and polyamides produced that have unusual properties compared to other thermoplastic polymers. Certain associative oligomers can vary from hard, clear, and low solubility ink jet resin to molecules that form gels with mineral oil and other low polarity compounds for candles and other consumer products, depending upon the composition. The amide bonds provide the oligomer association strength, while the variety of dimer structures limits crystal sizes to less than that needed for dispersion of visible light leading to clear products. For dimer polyamides, incorporation of medium chain bio-based diacids provide improved strength for the polyamides, while specific diamines provide a wide range of performance characteristics. The effect of the different diamines is explained by changes in hydrogen bonding between the different amide groups. These materials with a high bio-based content have been mainly ignored by current researchers working on bio-based materials.

## Figures and Tables

**Figure 1 polymers-15-03345-f001:**
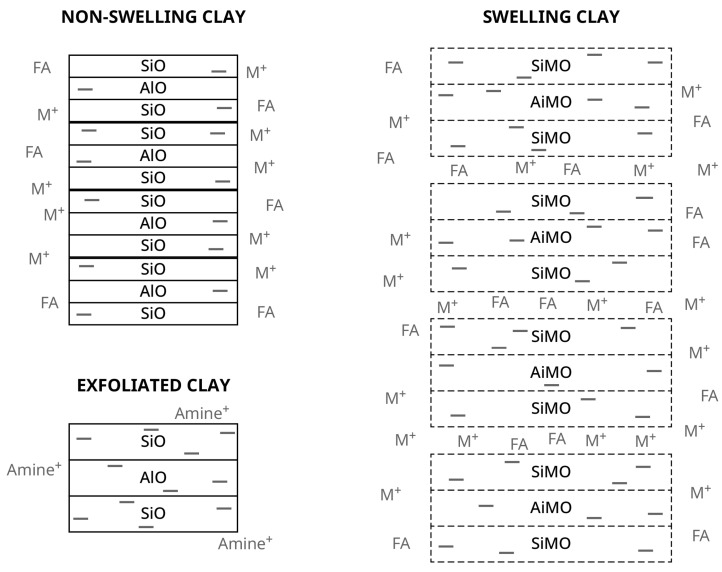
Structure of clays, showing charge and sheet separation differences for the non-swelling kaolinite on the left, the swelling smectite on the right and the exfoliated clay on the bottom left. SiO represents silicon oxide layer; AlO represents aluminum oxide layer: SiMO is the silicon oxide layer where some of the silicon atoms are replaced by lower valent metals creating a negative charge on the lattice; AlMO is the silicon oxide layer where some of the aluminum atoms are replaced by lower valent metals creating a negative charge on the lattice. Minus signs show negative charge on the lattices, M^+^ show counter balancing cations which can be mono-, di- or trivalent, and FA represents the adsorbed fatty acids.

**Figure 2 polymers-15-03345-f002:**
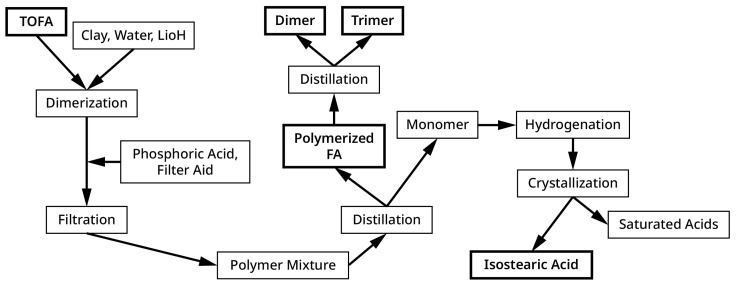
Schematic of the FA dimerization process and separation of products with the thicker bordered boxes representing commercial fatty acid products; often the hydrogenation step is skipped to yield a partially unsaturated and less expensive isostearic acid.

**Figure 3 polymers-15-03345-f003:**
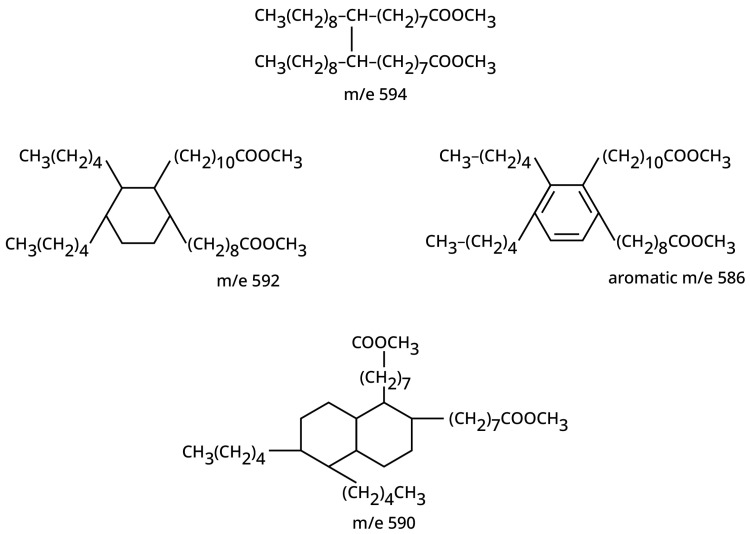
Likely structures of typical hydrogenated and esterified dimer molecules from the polymerization of unsaturated FA. Due to isomerization and minor FA monomers, the chain lengths and location of unsaturation of the attached chains vary, with the m/e from mass spectroscopy indicting the molecular weight. Reprinted with permission from McMahon and Crowell [38].

**Figure 4 polymers-15-03345-f004:**
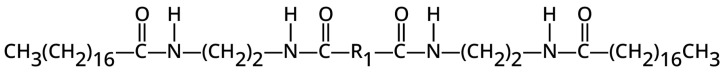
Desired composition of the oligomer is a dimer fatty acid (R_1_) with each end reacted with an ethylenediamine, that is then capped with a stearic acid. Figure taken from US patent 5,194,638 [45].

**Figure 5 polymers-15-03345-f005:**
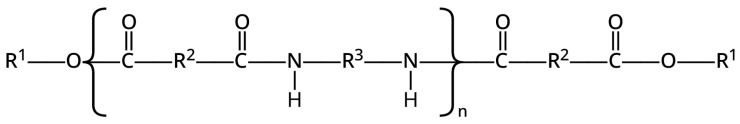
Structure for ester-terminated amides using a dimer core for gel structure and terminal ester groups for compatibility with non-polar components. Image taken from [47], with n being the number of repeat units of the oligomer, R^2^ being the 34 carbons of the dimer, R^3^ being short chain primary diamines and R^1^ being hydrocarbon groups with long chains.

**Figure 6 polymers-15-03345-f006:**
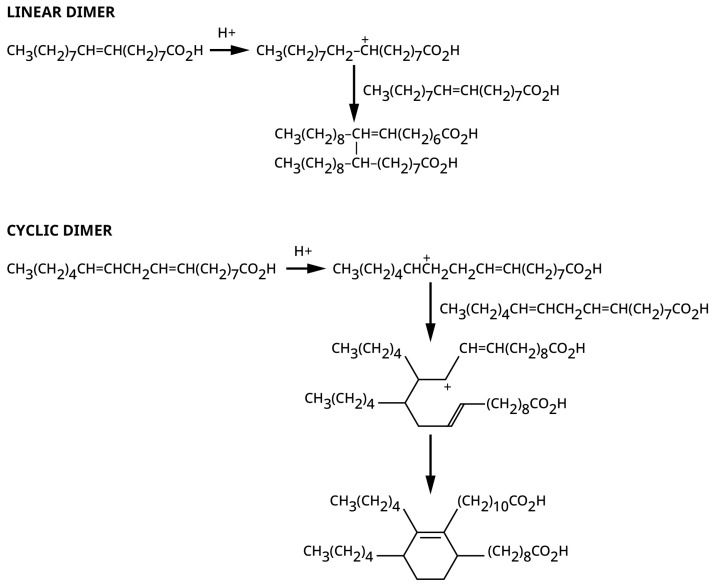
Typical type of cationic reactions to make linear and monocyclic dimers. These are only representative structures as the cationic reactions are generally not positional specific reactions due to isomerization of the unsaturated fatty acids, which in themselves have olefinic bonds at different locations.

**Figure 7 polymers-15-03345-f007:**
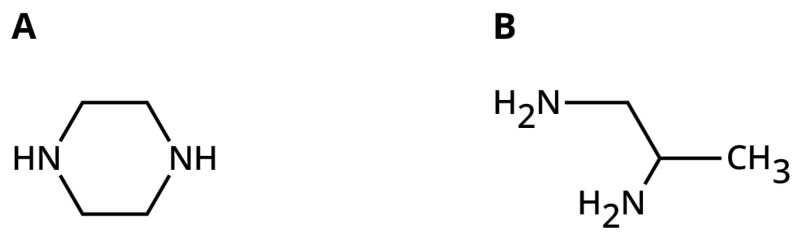
Left image (**A**) is of piperazine with only secondary amines. Right (**B**) is an image of 1,2-diaminopropane (DAP) with one hindered and one less hindered amine.

**Table 1 polymers-15-03345-t001:** Estimated structural composition of clay-catalyzed fatty acid dimers. Some methods, such as mass spectroscopy, do not readily provide quantitative data, and for other compounds, pure materials are not available for determining the response curves. Reprinted with permission from McMahon and Crowell [38].

Structure/FA Feed	Linoleic	TOFA	Oleic/Eladic
Linear (non-cyclic)	5	15 ^a^	40 ^a^
Monocyclic- aromatic	25	20	5
Monocyclic- nonaromatic	30	50 ^a^	50 ^a^
Polycyclic	40	15 ^a^	5 ^a^

^a^ Based on uncalibrated mass spectral data.

**Table 2 polymers-15-03345-t002:** Melting/softening point of various polyamides (nylons). Nylons have distinct melting points as determined by DSC, source: MP, TG, and Structure of Common Polymers (perkinelmer.com, (accessed on 5 June 2023)), while polyamides do not have distinct melting points, so ring and ball softening points are used [56].

Polymer	Monomers	Melting Point, °C	Softening Point, °C
Nylon 6	Caprolactam	210–220	
Nylon 6,6	Adipic acid, Heamethylene diamine	245–265	
Nylon 6,12	Dodecanoic acid, Heamethylene diamine	215–220	
Polyamide	Dimer acid, Ethylene diamine		100
Polyamide	Dimer acid, Heamethylene diamine		53–59
Polyamide	Dimer acid, Dimer diamine		Liquid

**Table 3 polymers-15-03345-t003:** Effect of co-diacid on the softening points, tensile strength, and elongation of dimer polyamides using ethylene diamine [24].

Dimer Acid, % of Total Acids	Sebacic Acid, % of Total Acids	Polyamide Softening Point, °C	Tensile Strength, MPa	Elongation, %
100	0	100	1600	500
95	5	135	1750	400
90	10	165	2100	350
85	15	200	3300	300

**Table 4 polymers-15-03345-t004:** Comparison of polyamides made with different short chain diamines using the same molar ratio of components, taken from the US patent 5,612,448 [68].

	Polyamide Adhesive Prepared from			
Physical Properties	1,2-Diaminopropane Example 1	Ethylenediamine Example 2	1,3-Diaminopropane Example 3	2-Methyl=1,5—Pentanediamine Example 4
Softening Point, °C	111	158	94	98
Viscosity at 190 °C, Pa-s	9.8	8.5	9.5	5.4
Tensile strength, MPa	2.48	3.61	0.52	0.09
Elongation, %	556	402	>2000	>800
Modulus, Pa	2335	12,690	*	*
Open time	50 s	10 s	>24 h	>24 h

* Too weak to measure accurately.

## Data Availability

No new data were created.

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
