# Peer review of "Chemistry of Dimer Acid Production from Fatty Acids and the Structure–Property Relationships of Polyamides Made from These Dimer Acids"

_polymers, 2023, doi:10.3390/polym15163345_

Round 1

Reviewer 1 Report

My main concerns for this manuscript are the use of data dating back to 1940s, with more than 36 references were taken from 19xx. I found this lack of sufficient recent data hindering the novelty of the work. I suggest authors to re-write the manuscript with more recent data, preferably within 10 years. For minor correction, authors need to revise the format of the manuscript such as subscription for chemical formulas. 

Only a few sentences need to be corrected.

Author Response

I thank you for reviewing the paper and making some important suggestions that I have answered in the revised draft.

Reviewer: My main concerns for this manuscript are the use of data dating back to 1940s, with more than 36 references were taken from 19xx. I found this lack of sufficient recent data hindering the novelty of the work. I suggest authors to re-write the manuscript with more recent data, preferably within 10 years. For minor correction, authors need to revise the format of the manuscript such as subscription for chemical formulas.

Response: I agree with the reviewer in that it would be better if there were more recent publications, but unfortunately the research community seems to be currently unaware of these biobased materials with very interesting chemistry and structure-property relationships. This lack of awareness despite a strong interest in the research community on biobased materials is the very justification for this paper that will hopefully provide researchers with sufficient information to inspire new research in this area that has good commercial track record instead of the umpteenth paper on lignin valorization without any scientific breakthrough or working on exotic by-products or processes that have no chance being commercially successful due to small volumes or the infrastructure needed for commercialization. Hopefully researchers will carry out the development of new products, carry out systematic research on the components to do mapping of performance versus composition, and apply techniques such as dynamic mechanic analysis versus temperature to better understand structure-property relationships.

 I have no idea what the reviewer means by a subscription for chemical formulas.

Reviewer 2 Report

Although this article is interesting, it requires major revision according to following lines:

1- Fig. 1 should be replaced with more professional image showing the swelling mechanism and intercalation/exfoliation mechanism.

2- Graphs from spectroscopic techniques to identify and prove the structure and chemistry of dimerization should be provided at section 3. More discussion on these results should be added (FTIR, HPLC, ...).

3- Dimerization mechanism should be represented in an image.

Author Response

I thank you for reviewing the paper and making some important suggestions that I have answered in the revised draft.

Although this article is interesting, it requires major revision according to following lines:

Reviewer: 1- Fig. 1 should be replaced with more professional image showing the swelling mechanism and intercalation/exfoliation mechanism.

Response: I tried to have the image simplified to make the differences clearer. I have seen many clay structure illustrations in the literature that are more detailed and professional, but they still fail to clearly illustrate the effect of cation substitution in the clay layers and how it influences the inter layer swelling. I have added more information in the text to try to relate structural changes to performance differences.

Reviewer: 2- Graphs from spectroscopic techniques to identify and prove the structure and chemistry of dimerization should be provided at section 3. More discussion on these results should be added (FTIR, HPLC, ...).

Response: The reviewer is correct in that I did not give enough discussion of the prior work. Since the paper by Crowell and McMahon does an excellent discussion on combining the results from various techniques and the literature to establish reasonable and unreasonable structures and mechanisms, I have added more discussion of this work and leave it to a reader to read the cited paper for a more in-depth discussion if they are interested.   

Reviewer 3 Report

Charles R. Frihart in his manuscript entitled " Chemistry of dimer acid production from fatty acids and the structure-property relationships of polyamides made from these dimer acids", reviewed possibilities to develop new adhesives from commercially available bio-based dimerized fatty acids. He suggested that the diacids derived from plant oils are valuable for making polyamides because of their very high bio-based content and highly tunable properties. However, some of the deficiencies listed below were observed:

  1. The manuscript contains some typos/spelling mistakes. They should be corrected.
  2. Since the dimerization process is considered the main process in this article, the author should expand Section 4 and try to explain this process to readers in more simple words.
  3. Tables and figures should be redacted according to MDPI style.
  4. DOI links for most references are absent. The author should provide them and rewrite all references according to MDPI style.
  5. Can the author provide the economic feasibility of the suggested adhesives based on bio-based dimerized fatty acids in comparison with those used in real-world applications?

Moderate editing of the English language is required.

Author Response

I thank you for reviewing the paper and making some important suggestions that I have answered in the revised draft.

Charles R. Frihart in his manuscript entitled " Chemistry of dimer acid production from fatty acids and the structure-property relationships of polyamides made from these dimer acids", reviewed possibilities to develop new adhesives from commercially available bio-based dimerized fatty acids. He suggested that the diacids derived from plant oils are valuable for making polyamides because of their very high bio-based content and highly tunable properties. However, some of the deficiencies listed below were observed:

 Reviewer: The manuscript contains some typos/spelling mistakes. They should be corrected.

Response: Without specific cases, I have reread the document and have had another person go over the paper to correct any errors that we could find.

Reviewer: Since the dimerization process is considered the main process in this article, the author should expand Section 4 and try to explain this process to readers in more simple words.

Response: I have added a figure and tried to clarify the explanation of the process and the dimer structure.

Reviewer: Tables and figures should be redacted according to MDPI style.

Response: Revisions have been made to the figures and tables to provide better consistency

Round 2

Reviewer 1 Report

Authors have satisfactorily revised the manuscript. It is now acceptable for publication.

Moderate revision is needed.

Reviewer 2 Report

It is acceptable.